# Diagnostic and Therapeutic Management of Primary Orbital Neuroendocrine Tumors (NETs): Systematic Literature Review and Clinical Case Presentation

**DOI:** 10.3390/biomedicines12020379

**Published:** 2024-02-06

**Authors:** Giulia Arrivi, Monia Specchia, Emanuela Pilozzi, Maria Rinzivillo, Damiano Caruso, Curzio Santangeli, Daniela Prosperi, Anna Maria Ascolese, Francesco Panzuto, Federica Mazzuca

**Affiliations:** 1Oncology Unit, Department of Clinical and Molecular Medicine, Sant’Andrea University Hospital, Sapienza University of Rome, Grottarossa Street 1035-1039, 00189 Rome, Italy; monia.specchia@uniroma1.it (M.S.); federica.mazzuca@uniroma1.it (F.M.); 2Department of Medical and Surgical Sciences and Translational Medicine, Faculty of Medicine and Psychology, PhD School in Translational Medicine and Oncology, Sapienza University of Rome, 00189 Rome, Italy; 3Anatomia Patologica Unit, Department of Clinical and Molecular Medicine, Sant’Andrea University Hospital, Sapienza University of Rome, 00189 Rome, Italy; emanuela.pilozzi@uniroma1.it; 4European Neuroendocrine Tumor Society (ENETS) Center of Excellence, Digestive Disease Unit, Sant’Andrea University Hospital, 00189 Rome, Italy; mariarinzivillo@gmail.com (M.R.); francesco.panzuto@uniroma1.it (F.P.); 5Radiology Unit, Department of Medical Surgical Sciences and Translational Medicine, Sant’Andrea University Hospital, Sapienza University of Rome, 00189 Rome, Italy; damiano.caruso@uniroma1.it (D.C.); curziosantangeli@gmail.com (C.S.); 6Nuclear Medicine Unit, Sant’Andrea University Hospital, 00189 Rome, Italy; dprosperi@ospedalesantandrea.it; 7Radiotherapy Oncology Unit, Department of Surgical Medical Sciences and Translational Medicine, Sant’Andrea University Hospital, Sapienza University of Rome, 00189 Rome, Italy; aascolese@ospedalesantandrea.it; 8European Neuroendocrine Tumor Society (ENETS) Center of Excellence, Department of Medical-Surgical Sciences and Translational Medicine, Sapienza University of Rome, 00189 Rome, Italy

**Keywords:** primary orbital, orbit, neuroendocrine tumor, NETs, multidisciplinary team, diagnosis, treatment

## Abstract

Background: The ocular involvement of neuroendocrine neoplasms (NENs) is uncommon and mainly represented by metastases from gastrointestinal and lung neuroendocrine tumors. Primary orbital NENs are even less common and their diagnostic and therapeutic management is a challenge. Methods: A systematic review of the literature was conducted from 1966 to September 2023 on PubMed to identify articles on orbital NENs and to summarize their clinical–pathological features, diagnosis and therapeutic management. Furthermore, we presented a case of a locally advanced retro-orbital primary neuroendocrine tumor that was referred to the certified Center of Excellence of Sant’Andrea Hospital, La Sapienza University of Rome, Italy. Results: The final analysis included 63 records on orbital NENs and 11 records focused on primary orbital NENs. The localization was mostly unilateral and in the right orbit; proptosis or exophthalmos represented the initial symptoms. The diagnostic work-up and therapeutic management was discussed and a diagnostic algorithm for the suspicion of primary orbital NENs was proposed. Conclusions: A multidisciplinary approach is required for the management of primary orbital NENs, emphasizing the importance of early referral to dedicated centers for prompt differential diagnosis, tailored treatment, and an improved quality of life and survival.

## 1. Introduction

Neuroendocrine tumors are rare neoplasms that arise from Kulchitsky, chromaffin, argyrophil, or argentaffin cells, which are distributed throughout the body but are mostly located in the gastrointestinal mucosa, lungs, biliary tract, pancreas, ovaries, testes, and thyroid gland [1]. Due to its indolent evolution, most diagnoses (50–75%) occur with metastatic disease, mainly in the lymph nodes, liver, and bones [2,3]. Instead, neuroendocrine cells are rarely detected in other organs, such as the brain or the organs of the senses, the eyes, or the orbital cavity. The incidence of brain metastases from neuroendocrine neoplasms (NENs) is reported to be between 1.5% and 5% [4] and is more frequent in primary lung neuroendocrine tumors (NETs); meanwhile, only 0.18% of brain disease comes from gastrointestinal NENs (GI-NENs), with a significant increase in the metastatic rate in poorly differentiated or undifferentiated NENs [2]. The eye and the orbital cavity are even more rare sites of cancer metastasis: approximately 8–10% of patients affected by a neoplasm of any histology or origin develop ocular metastases [5] in their cancer history; meanwhile, NENs account for 14–15% of all cases of orbital metastatic disease [6,7]. Although some reports in the literature are available [8,9,10,11], systematic epidemiological and clinical–pathological information about ocular metastases from NENs is still lacking. The most common site of the metastatic involvement of NENs is the uveal tract; these NENs arise from lung and breast NETs [8,12]; meanwhile, orbital cavity metastases are usually found in primary GI-NETs [13,14,15]. Only a few cases in the literature [16,17] have investigated neuroendocrine primary lesions of the orbit, suggesting that they are mostly represented by small-cell neuroendocrine carcinomas (SC-NECs) [18,19,20]. We report a locally advanced retro-orbital primary neuroendocrine tumor case that was referred to the European Neuroendocrine Tumor Society (ENETS)-certified Center of Excellence (CoE) of Sant’Andrea Hospital, La Sapienza University of Rome, Italy. Given the rarity of the orbital primary localization of neuroendocrine tumors, a comprehensive and systematic literature review was performed in order to systematically report all the clinical data available in the literature and outline a possible diagnostic algorithm and strategy for the therapeutic management of primary orbital NENs. 

## 2. Materials and Methods

We present the rare case of a 75-year-old woman diagnosed with a primary orbital NET, from diagnosis to treatment. A search of the literature, according to the Preferred Reporting Items for Systematic Reviews and Meta-analysis (PRISMA) guidelines [21] in PubMed, from 1966 to September 2023, was conducted to identify available articles on NENs of the orbit and their clinical management; retrospective studies, reviews, and case reports or case series restricted to the English language and human field were included. Comments or editorials and abstracts of congress were excluded from the analysis (PRISMA registration code: 490993). The search strategy was as follows: (“orbit” OR “orbital” OR “eye” OR “eyes”) AND (“Neuroendocrine Tumors” OR “Carcinoma, Neuroendocrine” OR “Carcinoid Tumor” OR “Neuroendocrine Tumor” OR “Tumor, Neuroendocrine” OR “Tumors, Neuroendocrine” OR “Neuroendocrine Carcinoma” OR “Neuroendocrine Carcinomas” OR “Carcinoid Tumors” OR “Tumor, Carcinoid” OR “Tumors, Carcinoid” OR “Carcinoid”). All the records were independently screened, via consultation and reading the abstracts, by two authors: G.A. and M.S. Other articles not comprising the initial search but relevant to our topic were included in our analysis. The PRISMA 2020 flow diagram for the search strategy was reported in Figure 1. Focusing on the primary neuroendocrine tumor of the orbit, the results regarding the demographic and clinical features, the diagnosis and treatment work-up are presented in tables and discussed. 

## 3. Case Presentation: Clinical History and Findings

In May 2020, a 75-year-old woman was referred to the ENETS-certified Center of Excellence (CoE) of Sant’Andrea Hospital, La Sapienza University of Rome, after several ophthalmic evaluations of a newly onset orbital mass of unexplained origin was carried out. She had been experiencing symptoms for three months: sporadic facial paresthesia, difficulty with facial movements, dysgeusia, and a visual acuity deficit in the right eye, which had worsened in recent months until complete blindness.

Before the diagnosis, the patient enjoyed good general conditions and led a regular life, and no relevant pathologies were reported in her pathological history. Initially, she was treated with steroid medications by a team of ophthalmologists and neurologists at another clinical center, but within a few months, she had not improved. For this reason, the patient underwent instrumental investigations using Computed Tomography (CT) scans and Magnetic Resonance Imaging (MRI), which demonstrated the presence of solid tissue that engaged the optic foramen, was located at the level of the right orbital cavity, and was in contact with the optic nerve and medial rectus muscle (Figure 2 and Figure 3).

MRI showed a hypointense mass in the baseline condition, with a subsequent contrast enhancement measuring 4 × 2.15 cm, growing through the right retrobulbar fat at the apex of the orbit; this involved the inferior orbital fissure and the lateral wall of the sphenoid sinus until it contacted the ipsilateral cavernous sinus. The lesion was arranged along the floor of the right orbit to infiltrate the inferior and medial rectus muscles and engaged the inferior orbital fissure in the context of the pterygopalatine fossa.

An incisional biopsy of the solid tissue in the pterygoid palatine fossa was performed. The tumor was composed of small to medium-sized cells with a hyperchromic nucleus and eosinophilic cytoplasm, with a mixed solid and trabecular growth pattern. Focal pseudo-glandular and papillary-like aspects with rosettes were present. The immunohistochemistry showed the intense and diffuse positivity of *CKAE1/AE3*, *S100*, *Synaptophysin*, *Chromogranin A*, *CD56*, *INSM1*, and *CDX2*, negativity for *CD99*, *Vimentin*, *CD20*, *CD3*, *p40*, *p62* smooth muscle actin, *TTF1*, *prolactin*, *ACTH*, *GH*, *FSH*, *LH, TSH*, *MELAN-A*, *CEA*, *p16*, and *p53*, and a Ki67 proliferation index of 6% (Figure 4). Focal areas of necrosis were observed with a mitotic rate of 0 mitoses/2 mm^2^; no figures of lympho-vascular and perineural invasion were reported. According to the 2019 WHO Classification [22,23], these findings suggested a diagnosis of well-differentiated neuroendocrine tumor (WD-NET G2), not excluding the possible diagnosis of ectopic hypophyseal adenoma (*S100+*) or a metastatic localization of gastroenteropancreatic neuroendocrine tumors (GEP-NETs) (*CDX2+*) [24]. Indeed, transcription factors can indicate the site of a primary tumor in the setting of an unknown primary. Although not entirely specific, lung origin is favored by *TTF-1*, pancreatic origin by *Isl1* and *PAX8*, and intestinal origin by *CDX2* [1,25]. 

Positron emission tomography/computed tomography (PET/CT) with 68-Gallium-DOTA-NOC revealed (68-Ga DOTA-NOC) an increased uptake in the right orbital cavity, a finding indicative of a neoplastic pathology with the high expression of somatostatin receptors (Figure 5).

The patient presented to us for further investigations and management and was referred to the Neuroendocrine Tumors Multidisciplinary Team (MDT) of our Institution. 

A baseline total-body CT scan was requested, which confirmed the findings of previous radiological exams, namely the contact of the solid tissue in the right orbital cavity with the optic nerve and the medial rectus muscle engaging the optic foramen. A histological review was carried out by an expert pathologist in the field of NEN features, considering the critical role of the central pathology review and the strict correlation between a valid pathological diagnosis and the pathologist’s expertise. The diagnosis of WD-NET G2 with a Ki67 proliferation index of 6% was confirmed. Immunohistochemical coloration showed *CDX2* positivity; immunostaining for pituitary transcription factors was performed to exclude the possible pituitary origin of the tumor. With the aim of excluding a midgut carcinoid, according to the suggestions of the pathological review and the literature data, which reported that most neuroendocrine orbital localizations are metastases of midgut NENs [13,14], CT enterography was performed; this showed no evidence of tumors in the small bowel. Moreover, a F-18 fluoro-deoxy-glucose (F-18 FDG) PET/TC revealed no FDG hyperaccumulation images in the right retro-orbital region and in the ipsilateral pterygoid palatine fossa (Figure 4).

According to these findings, the patient was started on treatment with somatostatin analogues (SSA)—long-term Lanreotide 120 mg—administered every four weeks. Given the symptoms of right eye blindness and exophthalmos, along with tinnitus and partial hearing loss, the patient was referred by MDT to a radiotherapist for palliative radiotherapy. Due to the difficult anatomical site of the lesion and the risk of consequent adverse effects, we decided to postpone an upfront radiotherapy treatment and to evaluate her clinical and radiological response to medical therapy. Surgery was not considered feasible after a multidisciplinary discussion due to the locally advanced aspect of the mass and the infiltrating involvement of nearby anatomical structures. 

Three years after diagnosis, radiological assessment—MRI and 68-Ga DOTA-NOC PET/CT—showed stable disease. The patient undergoes clinical follow-up visits every six months, and there have been marked improvements in her general condition, upward mobility of the right eye, exophthalmos, vision, and hearing. 

Written informed consent was obtained from the patient to participate in this study and for the publication of any potentially identifiable images or data included in this article.

## 4. Results

From a search of the literature with the aforementioned search strategy on PubMed, 348 records were identified. We excluded 275 records for irrelevant content after they were screened by two authors independently. Of the remaining 73 potentially eligible studies, one duplicate, one comment, one letter to the Editor, two with insufficient information, and five in a non-English language were excluded (Figure 1).

We included, in the final analysis, 63 records on orbital neuroendocrine tumors published between 1966 and September 2023, which comprised 52 case reports or case series, 5 reviews, and 6 retrospective studies (Table 1). In the majority of cases, the gastrointestinal tract was the primary tumor’s location (*n* = 38 records), while the respiratory tract (*n* = 16 records) was the second most represented location. Furthermore, although infrequent, neuroendocrine orbital metastases originating from kidney (*n* = 6), ovary or testicular (*n* = 3), and breast (*n* = 1) primary tumors were described. Thirteen articles did not specify the primary site of tumor or reported tumors of “unknown origin”.

Table 2 lists all the studies investigating exclusively ocular primary NETs that were reported in the literature. Including our report, a total of 16 patients presented with primary orbital NENs, including nine females and seven males, with a median age of 56.5 years (range 25–82) at diagnosis. The laterality of ocular involvement was right and left in 11 and 5 cases, respectively, and the initial symptoms were mostly represented by proptosis or exophthalmos (*n* = 11.69%), ocular pain (*n* = 3.19%) and the loss of or reduction/alteration in vision (*n* = 5.31%). Neurological symptoms [26,27] and Cushing’s syndrome [28,29] were described in only two cases, respectively. The histological diagnosis was available for all patients, according to the histopathological classification used at the time of publication of the different cases: “carcinoid” was described in five cases, well-differentiated NETs in two, and NEC/poorly differentiated carcinoma/small-cell NEC in the remaining nine.

In all patients, the mass was diagnosed with an MRI or CT scan; a visual acuity exam was performed in the seven patients with loss or blurred vision. The use of 68-Ga and (F-18) FDG PET/CT was reported only in two case reports, including our patient [18]. Surgery represented the backbone of treatment in twelve patients, of whom five also received post-operative radiotherapy and two received chemotherapy. In one case, it was proposed that CT and RT treatment be combined [18], while two patients, including ours, were treated with exclusive medical treatment [30]. No data about the therapeutic algorithm proposed was available in one report.

**Table 1 biomedicines-12-00379-t001:** Literature data on orbital NETs published from 1966 to September 2023.

First Author, Year of Publication	Type of the Study	Patients (*n*)	Age or Median Age	Site of Primary Tumor	Histotypes of Primary Tumor (WHO)	Treatment	Outcomes
Font, R.L., 1966 [31]	case report	1	74	Lung	NEN	Surgery	NA
Rush, J.A., 1980 [32]	case report	1	58	Colon	NEN	Surgery, radiotherapy	NA
Harris, A.L., 1981 [33]	case report	1	NA	NA	NA	NA	NA
Riddle, P.J., 1982 [17]	retrospective	15	53	LUNG (53.3%), GI (26.6%), NA (20%)	NETs	Surgery (80%), Chemotherapy (6.6%), Radiation (13.3)	OS 5.2 y
Zimmerman, L.E., 1983 [16]	case report	1	71	Orbit	Mixture of the histologic patterns	NA	NA
Shields, C.L., 1987 [15]	case report	1	63	Ileum	NEN	Surgery	NA
Harris, G.J., 1989 [34]	review	NA	NA	GI	NA	NA	NA
Shetlar, D.J., 1990 [8]	case report	3	53.3	GI (33.3%); Liver (33.3%); Unknown (33.3%)	NETs (66.6%); Unknown (33.3%)	Surgery (100%)	OS 4 y (33.3%), PFS 7.5y (33.3%); NA (33.3%)
Nida, T.Y., 1992 [35]	case report	1	63	Respiratory Tract	NEC	surgery	NA
Aburn, N.S., 1995 [36]	case report	1	64	Lung	AC	Local radiotherapy, chemotherapy	OS 1 y
Fan, J.T., 1995 [9]	case series	7	59	Bronchus (42.8%); Ileum (28.6%); Colon (14.3%); Unknown (14.3%)	NENs (100%)	Iodine 125 brachytherapy (28.6%); Chemotherapy (71.4%); Ruthenium brachytherapy (14.3%); Enucleation (14.3%); External beam irradiation (42.8%)	FU 7–8 y (57.1%); FU 1–2 y (42.9%)
El-Toukhy, E., 1996 [37]	case report	1	45	GI	NET	Surgery, Radiotherapy	NA
Hanson, M.W., 1998 [38]	case report	1	76	Ileum	Carcinoid tumor from Kulchitsky cells	NA	NA
Couch, D.A., 2000 [39]	case report	2	50	Pancreas; ileocecal tumour	Carcinoid tumor	Radiotherapy; Surgery	OS 2 mo; disappearance of symptoms after surgery
Khaw, P., 2001 [40]	case report	2	58	Mediastinum (50%); Kidney (50%)	AC (50%), NEN (50%)	Radiotherapy (100%)	No symptoms after treatment
Takemoto, Y., 2003 [41]	case report	1	39	Duodenum	NEN	Surgery	NA
Sivagnanavel, V., 2004 [42]	case report	1	70	NA	NET	Ocreotide therapy,	NA
Borota, O.C., 2005 [43]	case report	1	61	GI	NEN	Surgery, Interferon/Sandostatin	NA
Mehta, J.S., 2005 [11]	case series	13	65.3	Ileal (53.8%); Colon (30.7%); Bronchus (7.6%); Breast (7.6%)	NEN	Exenteration (30.7%); Radiotherapy after tumor debulking (38.4%); Radiotherapy alone (15.3%); Local radiotherapy with receptor-targeted chemotherapy (15.3%).	5 y OS 72%
Kiratli, H., 2008 [44]	case report	1	74	Lung	AC	Cisplatin/Etoposide	NA
Mititelu, M., 2008 [23]	case report	1	60	Orbit	NET	Chemotherapy	OS 4 y
Bohman, E., 2009 [45]	case report	1	69	Midgut	NEN	Sandostatin	NA
Matsuo, T., 2010 [46]	case report	1	72	Retroperitoneum	WD-NET	Radiation	NA
Sira, M., 2010 [7]	case report	1	78	NA	NET	Octreotide therapy	OS 1 y
Gupta, A., 2011 [47]	case report	4	61.9	Pancreas (14.3%); Unknown (85.7%)	NETs (100%)	Chemotherapy (14.3%), Unknown (85.7%)	NA
Turaka, K., 2011 [48]	case series	4	58–66	GI (50%); lung (25%); testicle (25%).	NETs	Surgery (75%); external beam radiotherapy (50%)	FU 39 mo
Yang, M., 2011 [49]	case series	3	63	GI (33.3%), NA (66.6%)	NETs (66.6%), NEC (33.3%)	Surgery (100%), Radiotherapy (66.6%)	FU 3.5 y
Mehta, P., 2012 [50]	case series	2	50	Unspecified, with bone metastases; ileal carcinoid tumour	NENs	Octreotide therapy	In the first case, it was necessary to use occlusive lenses to control diplopia; in the second case, complete resolution of ptosis after treatment
Atik, A., 2013 [20]	case report	1	35	Right orbit	NEC	Surgery, Chemotherapy and Radiotherapy	OS 2 y
Dobson, R., 2013 [51]	case report	1	65	Midgut	NET	PRRT using Lu-DOTATATE	FU 14 mo
Peixoto, R.D., 2013 [52]	case report	1	79	GI	NET	Octreotide therapy, radiotherapy and Capecitabine/Temozolamide	NA
Kiratli, H., 2015 [53]	case report	1	57	Kidney	NET	Octreotide therapy, Capecitabine/Temozolomide	FU 24 mo
MacIntosh, P.W., 2015 [54]	review	1	35	Antrum and Orbit	SNEC	Surgery, chemotherapy and radiotherapy	NA
Parikh, D., 2015 [55]	case report	1	70	Kidney	NEN	Octreotide therapy, everolimus	FU 6 mo (stable proptosis without diplopia)
Makis, W., 2016 [56]	retrospective	4	67	GI (75%), LUNG (25%)	NETs (100%)	Lu-DOTATATE (50%); I-MIBG (50%)	PFS 27.75 mo
Matějka, V.M., 2016 [57]	review	1	50	Appendix	GCC	Surgery, Chemotherapy	Death a few months after diagnosis
Ruiz, J.B., 2016 [58]	case report	1	53	Thyroid	medullary thyroid carcinoma	Surgery, External radiation	4 mo later: morphological stability and reduction in calcitonin levels.
Warren, C.C., 2016 [59]	case report	1	70	GI	Carcinoid tumor	Resection, chemotherapy	NA
Kamaleshwaran, K.K., 2017 [60]	case report	1	45	Respiratory Tract	NET	LU_DOTATATE	NA
Chen, Y.J., 2018 [61]	case report	1	69	Pancreas	NET	Surgery, Octreotide therapy	NA
Das, S., 2018 [62]	case series	5	67.2	Ileal (100%)	NETs	Chemotherapy (20%); Radiation Therapy (IGRT) (40%;) 177Lu-DOTATATE (PRRT) (20%); surveillance (20%)	NA
Guo, Y., 2018 [63]	case report	1	86	Bronchial	NEN	Octreotide acetate, dacarbazine, carboplatin, etoposide	NA
Rasmussen, J.Ø., 2018 [64]	case report	1	71	Liver	NEC	Surgery, radiotherapy, IntronA/Sandostatin/Nexavar/Dotatoc, Temodal, etoposide monotherapy	FU 15 y
Thyparampil, P.J., 2018 [24]	case report	1	64	Orbit	NET	Surgery	NA
Beam, A.S., 2019 [65]	case report	1	86	GI	NET	Octreotide therapy	NA
Hafiz, A.A., 2019 [66]	case report	1	50	Small bowel	NET	Surgery. Octreotide therapy	NA
Kozubowska, K., 2019 [67]	case report	1	73	GI	NET	Surgery	NA
Mittal, R., 2019 [19]	case report	1	25	Left orbit	NEC	Chemotherapy	OS 21 mo
Parghane, R.V., 2019 [68]	case report	1	53	Kidney	NET	Radiotherapy, Octreotide therapy, (Lu)—DOTATATE PRRT	PFS 18 mo
Bacorn, C., 2020 [69]	case report	1	65	GI	NET	Radiotherapy, octreotide therapy	FU 1 y
Kamieniarz, L., 2020 [70]	retrospective	27	53	Small bowel (66.4%); Pancreas (14.8%). Hepatic with or without concomitant skeletal metastases (85%)	NEN	External bema radiotherapy (45.4%)	5 y OS 84.1%
La Salvia, A., 2020 [13]	review	64	NA	Lung (43.7%); Iileum (21.8%); Unknown (21.8%); Colon (4.6%); Rectum (1.5%); Esophagus (1.5%); Thymus (1.5%); Testicle (1.5%); Liver (1.5%)	NETs (39%)	Surgery (7.8%); External bema radiotherapy (29.6%); brachytherapy (7.8%); Photocoaugulation (1.5%)	NA
Mustak, H., 2021 [71]	observational cross-sectional cohort, retrospective	28	58.8	Gastrointestinal tract (GI) (50%); lung (7.1%); kidney (7.1%) and ovary (3.6%)	NETs	Observation (14.3%); Radiation (35.7%); Octreotide (39.3%); Chemotherapy (32.1%); Steroids (10.7%); Surgery (60.7%)	5 y OS 81.8%
Albanese, G., 2022 [14]	case report and review	95	63.8	GI (62–85%)	NETs		NA
Isidori, A.M., 2002 [72]	case series	6	52.8	Bronchial (50%), Ileum (16.6%), Unknown (33.3%)	NENs (83.3%), NA (16.6%)	External Beam Radiotherapy (16.6%), 200 mCi I[MIBG] (16.6%); Surgery (16.6%); Radiotherapy (33.3%); Chemotherapy (50%)	OS 1 y (16.6%), FU 3 y (16.6%); FU 6 y (16.6%); NA (50%)
Matsuo, S., 2022 [73]	case report	1	71	GI	NET	Surgery	NA
Meinhardt A, 2022 [74]	case report	1	47	Extrapulmonary	NEC	Cisplatin/Etoposide, radiotherapy	OS 1–2 y
Morland, D., 2022 [75]	retrospective	4	NA	GI (100%)	NETs (100%)	NA	NA
Ryan, T.G., 2022 [76]	retrospective	9	53	Bowel (33%), kidney (22%) pancreas (11%)	NETs	Chemotherapy and Radiotherapy	Died after diagnosis
Shen, A., 2022 [10]	case series	2	29.5	GI (100%)	NET (50%), SNEC (50%)	Capecitabine/Temozolamide (50%); Cisplatin/Etoposide, radiotherapy, Topotecan, Paclitaxel and Nivolumab (50%)	NA
Agrawal, S., 2023 [18]	case report	1	33	Respiratory Tract	SNEC	Chemotherapy	NA
Woods, M.D., 2023 [77]	case report	1	79	GI	carcinoid	Somatostatin analogue	OS 6 y
Zhang, Y.S., 2023 [27]	case report	1	70	Without a systemic primary site	NEC	Chemotherapy	OS 1 mo

Abbreviations: NETs: neuroendocrine tumors; WD: well differentiated; NEC: neuroendocrine carcinoma; NENs: neuroendocrine neoplasms; SNEC: small-cell neuroendocrine carcinoma; GCC: global cell carcinoid; AC: atypical carcinoid; PRRT: peptide receptor radiolabeled therapy; NOS: no other specified; GI: gastrointestinal; OS: overall survival; y: year; mo: months; NA: not available; FU: follow-up; PFS: progression free survival

**Table 2 biomedicines-12-00379-t002:** Characteristics of patients with ocular primary NETs analyzed in the present study.

Study, Year	Age/Sex	Clinical Presentation	Exams at Diagnosis	Histotypes (WHO) *	Histological Pattern	Laterality	Treatments	Outcomes
Zimmerman et al., 1983 [16]	71/F	Exophthalmos; loss of vision	5-HIAA levels; CT scan	Carcinoid 1/70 HPF	Mixture of basaloid, trabecular, tubular and rosetted histologic patterns, positive argentaffin reaction, foci of PASS positive; Neurosecretory granules	Right Orbit	NA	NA
El-Toukhy et al., 1996 [37]	45/M	Proptosis, increased tearing, painless swelling	Visual field, CT scan; arteriogram	Carcinoid	Positive argentaffin and argyrophilic reactions	Right Orbit	Surgery; radiation	1976 first orbital tumor; 1979 orbital relapse; 1982 ileal carcinoid; 1986 orbital tumor (pt alive in 1996) OS 20 y
Mititelu et al., 2008 [26]	60/M	Cranial neuropathies, pupillary abnormalities, ptosis, ocular motility disturbance	MRI brain, MRI venography, MRI angiography, cerebral angiogram, CT chest/abdomen, cerebrospinal fluid cytology	Poorly differentiated neuroendocrine carcinoma	Synaptophysin, positive perinuclear distribution, negative for Cam 5.2, chromogranin, CD56, neuron-specific enolase, S100 protein, thyroid transcription factor-1	Left Orbit	Surgery and CT (CBDCA VP-16)	OS 4 y (dead for meningitis spread)
Yang et al., 2011 [49]	54/M	Proptosis, loss of vision, abdominal pain	MRI imaging; best-corrected visual acuity (BCVA)	Low grade carcinoid	Ovoid monomorphic cells with “salt and pepper” chromatin and nucleoli	Right Orbit	Surgery	NA
	59/M	Proptosis	BVCA, MRI	Poorly differentiated carcinoma	Epithelioid cells with pleomorphic nuclei	Right Orbit	Surgery + adjuvant radiotherapy (60 Gy in 30 fractions	OS 5 y
	78/F **	Asyntomatic	BVCA, MRI	Poorly differentiated	Cells with hyperchromatic, angulated, nuclei and scanty cytoplasm. Mitotic figures were readily found.	Right Orbit	Surgery + radiotherapy	OS 3.5 y **
Atik et al., 2013 [20]	35/F	Periorbital pain, progressive swelling and blurred vision	Visual acuity exam, MRI, CT scan, PET	Small-cell neuroendocrine carcinoma (SCNEC)	Small blue cells arranged in broad sheets, nests and cords	Right Orbit	CT (CCDP 5FU) followed by surgery, adjuvant CT CDDP-based + RT	OS 2 y
Thyparampil et al., 2018 [27]	64/F	Proptosis, orbital pain, eyelid swelling after several weeks of ipsilateral headaches	MRI, chest/abdomen/pelvis CT scan, octreoscan	WD-NET (KI67 3–5%)	Cells with abundant cytoplasm, fine granular (salt and pepper) chromatin, rare mitoses	Right Orbit	Surgery	OS 1 y
Mittal et al., 2019 [19]	25/F	Pain, proptosis, swelling	Visual acuity exam, CT scan	Small-cell neuroendocrine carcinoma (Ki67 90%)	Small round blue cells with scant cytoplasm, and dense nuclei amidst necrotic debris	Left Orbit	Surgery + CT (CDDP VP-16) +RT	OS 21 mo
Tan, H. et al., 2020 [29]	48/F	Cushing’s syndrome	ACTH test, MRI	Typical Carcinoid	positive staining of ACTH, chromogranin A, Syn, PCK and EMA	Left Orbit	Surgery	NA
Agrawal et al., 2023 [18]	33/M	Proptosis, swelling	PET-CT Ga-68, PET-CT 18-F, MRI	High grade SCNEC	Small round blue cells with scanty intervening stroma, abundant necrosis and apoptosis	Right Orbit	CT 6 cycles + RT	NA
Jun, Y.Q., 2023 [28]	48/M	Proptosis	Abdominal CT and bone marrow evaluation	Carcinoma (Ki67 80%)	Poorly differentiated	Right Orbit	Surgery	OS 3 mo
	82/F	Progressive mass on the lower eyelid	Total body CT, bone marrow evaluation	Carcinoma (Ki67 90%)	Poorly differentiated	Left Orbit	Surgery	OS 3 mo
	48/F	Cushing syndrome	Cortisol/ACTH levels, pituitary/orbital MRI, abdominal CT	Carcinoid	Carcinoid	Left Orbit	Surgery	OS 4 y
Zhang, Y.S., 2023 [30]	70/M	painless bilateral eyelid oedema and vertical diplopia	MRI, chest X-ray, Total body CT, PET-CT 18-F	Carcinoma (Ki67 70%)	Poorly differentiated high-grade neuroendocrine carcinoma	Right Orbit	CT (CBDCA VP-16)	OS 1 mo
Arrivi et al., 2023 (our case)	77/F	Proptosis, loss of vision	PET-CT Ga-68 PET-CT 18-F, MRI, total body CT, CT Enterography Visual acuity exam	WD-NET G2 (Ki-67 6%)	WD-NET	Right Orbit	Lanreotide	OS 3 y

Abbreviations: F: female; M: male; CT scan: Computed Tomography scan; OS: overall survival; NA: not available; HPF: high-power fields; 5-HIAA: 5-hydroxyindoleacetic acid; MRI: magnetic resonance imaging; pt/pts: patient/patients; Cam 5.2: low molecular weight cytokeratin; CT: chemotherapy; RT: radiotherapy; CBDCA: carboplatin; VP-16: etoposide; PET-CT: positron emission tomography/computed tomography; CCDP: cisplatin; 5FU:5 fluorouracil; WD-NET: well-differentiated neuroendocrine neoplasia; G: grading. * WHO Classification was reported according to paper’s publication year. ** pt developed a colon neuroendocrine tumor one-year post orbitotomy.

## 5. Discussion

In the natural history of neoplasms of any origin and histological type, ocular involvement is a rare occurrence. This finding is even more uncommon in NENs, which account for about 14–15% of all orbital metastatic disease [6,7], mainly involving the uvea, the orbital cavity, and the extraocular muscles [13,14,78].

Ocular metastases, with no substantial gender predominance, occur mostly in GI-NENs (range 62–69%) and in the lung (range 7–23%), but primary testicular/ovarian, breast, and renal tumors are also described in the literature [13,14].

Although several reviews and cases in the literature have been published about neuroendocrine orbital metastases [13,14,34,57], the management of NENs ocular metastases still appears difficult and not well defined by algorithms and guidelines. Even more unexplored is the issue of primary orbital NENs, and according to our knowledge, this is the first systematic review to focus on primary orbital NENs and present a proposal for diagnostic and therapeutic work-up in the ENETs CoE.

Only 15 cases of primary orbital NENs, excluding our case, have been described in the literature since the 1980s. The median age at diagnosis was 56.5 years (range 25–82), with a slight prevalence of female sex and right orbit involvement. Unlike what was observed for metastatic sites—12.5% of metastases are bilateral [13]—no ocular bilateral involvement was described in primary tumors. Only one case showed an initial involvement of the right eye, a subsequent involvement of homolateral cervical nodes, and further disease progression with tumor spread in the contralateral left orbit [20].

The clinical presentation is often characterized by a progressive orbital mass with proptosis, swelling, and visual disturbances such as diplopia, unilateral blindness, and blurry vision, while only a few cases are associated with pain and neurological symptoms such as headaches or neuropathy [26,27]. A small number of orbital primary neoplasms can secrete hormones, leading to carcinoid syndrome with the ectopic release of adrenocorticotropic hormone (ACTH) [28,29]. Therefore, although Cushing’s Syndrome is mainly associated with neuroendocrine lung cancers [79,80], it should be taken account of as an uncommon but described clinical occurrence of this rare NEN localization. The diagnosis of Cushing’s syndrome is important but challenging, and a complete physical examination to identify alopecia, skin pigmentation, moon face, central obesity, multiple purple striae on the abdomen, scattered bruises, and proximal muscle weakness could help the physician achieve a timely differential diagnosis. We propose that the algorithm used to diagnose Cushing’s Syndrome is proceeded with in cases of suggestive symptoms [81].

The 5-hydroxyindolacetic acid (5-HIAA) urinary levels of patients should be evaluated in the presence of chronic diarrhea, flushing, and/or bronchospasm, even though the classic form of carcinoid syndrome was not described in orbital NENs; however, it was predominantly encountered in patients with WD-NENs of intestinal origin and lung NETs [82].

Blood tests and neurological examinations with assessments of peripheral and central nerve function can be useful to exclude endocrinological or neurological disorders. In cases of visual disturbances, an ophthalmologist consultation with visual acuity and visual field tests should be pursued. Given the limited accuracy of physical or ophthalmologic examination findings for a differential diagnosis, facial and brain MRI or CT scans should be mandatory in the diagnostic work-up of orbital NENs. Metastatic lesions appear as well-circumscribed, fusiform, or round masses in extra-ocular muscle, do not show uniform hypervascularity on MRI and have both homogeneous and heterogeneous density on CT scans [47,76]. Instead, primary orbital NENs show highly variable imaging characteristics such as intense enhancement [18,20,29] or heterogeneous aspects, with patchy or mild enhancement on MRI [26,49]. It must be considered that MRI is superior to CT for the imaging of the bones and the brain [83,84]; therefore, it may be advisable to use MRI when orbital NENs are suspected. To exclude a different primary site than the orbit, a CT scan of the abdomen and thorax is suggested. Given that the majority of orbital localizations originate from primary intestinal NENs [13,70], the use of CT enterography (CTE) or MR enterography (MRE) can be useful to detect small bowel lesions, especially if the signs and symptoms of carcinoid syndrome are noted [85,86]. El-Toukhy described a case of orbital carcinoid with several local recurrences that was diagnosed six years before the detection of ileal NEN. This case suggests the possibility that the orbital NEN was not the primary neoplasm, but rather the metastasis of an occult small intestine tumor [37]. For this reason, we advise integrating CTE, MRE or functional methods in the diagnostic work-up of these NENs from the first diagnosis.

The sensitivity required to detect NENs by 68-Ga somatostatin analogs PET/CT is 92% (range 64–100%) and the specificity is 95% (range 83–100%) [87]. Indeed, 68-Ga PET/CT could help to detect additional or unknown lesions compared to conventional radiological methods such as CT or MRI [62,83,88] and play a central role in therapy selection, identifying patients eligible for peptide receptor radionuclide therapy and monitoring their response to treatment [89,90]. The use of (F-18) FDG PET/CT, due to its prognostic role and according to guidelines [84], is optional and suggested in G2/G3 NETs, which generally have a higher glucose metabolism and a lower somatostatin receptor expression [91,92,93]; meanwhile, it represents the main functional method used for investigation in the study of NECs [83]. When using functional imaging techniques in the diagnostic work-up of orbital NENs, we must consider that somatostatin receptors are expressed in several normal human tissues, including the brain [94,95,96], and that (F-18) FDG is accumulated in the brain with a very high uptake. Despite the use of PET/CT, as recommended by European guidelines from 2017 [83], only three patients, including our case [18,30], underwent functional studies, and only two underwent double-tracer PET/CT.

The histopathological analysis of tissue samples remains the gold standard for diagnosis and can be performed through incisional biopsy or the surgery of the whole lesion if resectable. The histomorphological growth pattern and cytology evidenced on hematoxylin eosin (HE)-stained tissue suggest the diagnosis of NENs, but the neuroendocrine phenotype is proven by the immunohistochemical detection of the neuroendocrine markers synaptophysin and chromogranin A [84]. A comprehensive immunohistochemical analysis for transcription factors is very helpful in the identification of tumor differentiation and the site of origin of a metastatic tumor [97]. Furthermore, considering the strict correlation between the correct pathological diagnosis of NENs and the pathologist’s expertise, a central pathology review is always recommended when feasible. The most represented pathologic features of primary orbital neoplasms are those of poorly differentiated neuroendocrine carcinomas (*n* = 9); meanwhile, only two cases were described as well-differentiated NETs, and the other five cases were reported with a generic term of “carcinoid”. Over time, there has been an evolution of the nomenclature in NEN classification [97], so we must carefully consider the histopathological definitions of the cases, especially due to the different times of publication and the lack of a central pathology review. Despite the low strength of evidence in the literature in this field, a proposal of the diagnostic algorithm to be used if a primary orbital NEN is suspected is presented in Figure 6.

The combination of histopathological characteristics and functional imaging represents the possibility of acquiring a more comprehensive understanding of individual patients’ disease biology and prognostic aspects, leading to a proper diagnostic and therapeutic algorithm. In this field, a dedicated MDT appears crucial in ensuring the quality of the clinical management of NENs [98]. Therapeutic decision making should always be discussed with the components of the MDT, and even more so in these rare primary site tumors.

The backbone of therapeutic management has been surgery since the first cases were reported in 1996; meanwhile, subsequent or concomitant treatments have been proposed according to histology. Surgery should be always offered in resectable disease, irrespective of histology, although retrospective series indicate that it is rarely curative alone in NECs [99]. Given the high risk of relapse, adjuvant platinum-based chemotherapy or radiotherapy could be considered. Radiotherapy should also be proposed in localized unresectable disease with or without chemotherapy, or lastly as palliative treatment for the control of symptoms. Although some patients have been treated with cytoreductive chemotherapy [18,20], currently, there are insufficient data to recommend this treatment. The management of advanced disease is currently based on evidence in the literature regarding gastrointestinal and lung NECs [100,101,102,103]; consequently, cisplatin or carboplatin plus etoposide represented the most commonly used chemotherapy regimen. Evidence for salvage therapy in patients progressing on first-line platinum-based chemotherapy is still lacking regarding orbital NECs; only in one case of relapse was intra-arterial chemotherapy used, with disease progression after three months [20]. In metastatic WD-NETs, the therapeutic decision making should be essentially based on the grading, functionality, somatostatin receptor status, tumor extent, and hepatic tumor burden if metastatic disease, and should refer to consensus guidelines [84,99,104].

The median OS of primary orbital NENs was 2.5 years (range: 1 month—20 years), and the outcome was not reported in four cases. This is except for one patient who presented with leptomeningeal spreading four years after the first diagnosis; the typical attitude of these neoplasms is of locoregional involvement with no distant metastases described. Currently, we are unable to produce a clear recommendation regarding follow up-management, but we suggest that clinical and radiological evaluations with MRI and PET/CT are performed for at least 5 years, according to national and international oncological guidelines [84,105,106].

## 6. Conclusions

Primary orbital NENs represent a rare disease, and the lack of comprehensive epidemiological and clinical–pathological data makes their management difficult. Due to this complexity, a multidisciplinary approach is widely encouraged in order to guarantee the best diagnostic and therapeutic work-up for our patients [107]. In this context, this review aims to be useful as a guide for our colleagues who are dealing with primary orbital neuroendocrine tumors, summarizing the most typical symptoms at diagnosis, with a focus on the diagnostic examinations that must be performed to promptly differentiate between orbital primary neoplasms and metastases.

Our case emphasizes the importance of early referral to NEN-dedicated centers and the important role of the MDT in prompt differential diagnosis; this would provide the patient with tailored treatment and an improvement in their quality of life and survival [108,109].

## Figures and Tables

**Figure 1 biomedicines-12-00379-f001:**
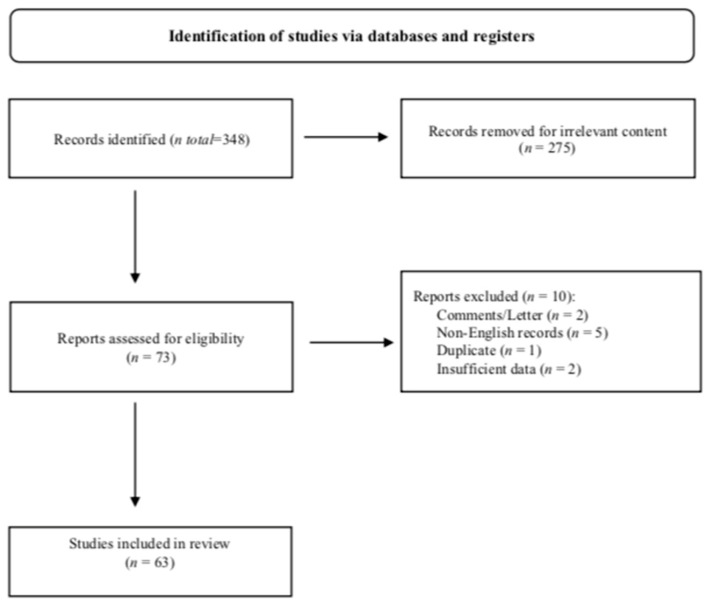
PRISMA 2020 flow diagram for the search strategy.

**Figure 2 biomedicines-12-00379-f002:**
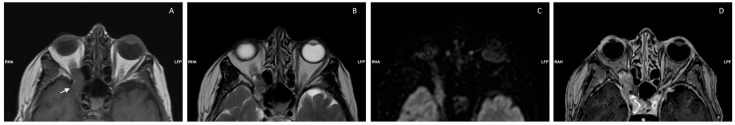
MRI sequences above show pathologic tissue (arrow) growing through the right retrobulbar fat at the apex of the orbit; this involves the inferior orbital fissure and the lateral wall of sphenoid sinus, until it contacts the ipsilateral cavernous sinus. (**A**) Axial T1 FSE: the lesion appears iso-intense to muscle. (**B**) Axial T2 weighted: the lesion shows a non-homogeneous signal intensity due to areas of cystic degeneration in the ventral part. (**C**) DWI: the lesion shows a restriction of the proton density. (**D**) Fat-saturated contrast-enhanced axial T1 weighted: the lesion shows high enhancement.

**Figure 3 biomedicines-12-00379-f003:**
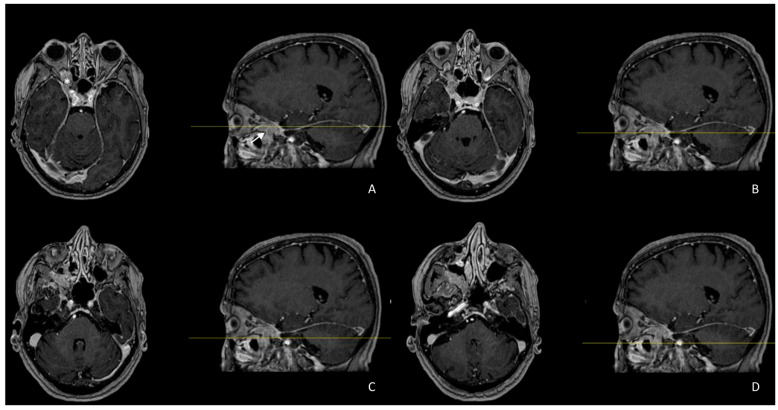
(**A**–**D**) Fat-saturated contrast-enhanced T1-weighted sequences, both axial and sagittal show, respectively, the lateral (*) and longitudinal (arrow) extension of the lesion. The pathological tissue is arranged along the floor of the right orbit to infiltrate the inferior and medial rectus muscles; this engages the inferior orbital fissure into the context of the pterygopalatine fossa.

**Figure 4 biomedicines-12-00379-f004:**
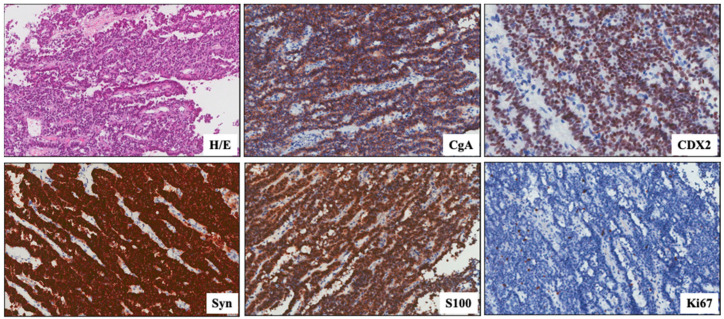
Photograph of orbital biopsy specimen. Hematoxylin and eosin (HE) stain of paraffin-embedded tumor tissue and immunochemistry with antibodies specific for *Chromogranin A (CgA)*, *CDX2*, *synaptophysin (Syn)*, *S100*; Ki67.

**Figure 5 biomedicines-12-00379-f005:**
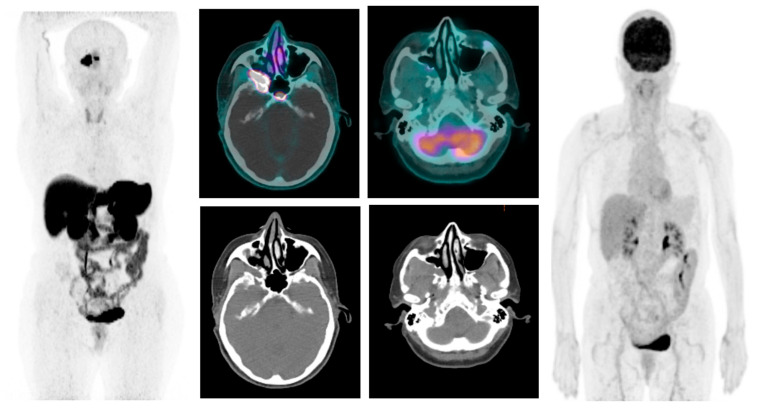
Baseline images of 68Ga-DOTA-NOC PET/CT (**left**) and (F-18) FDG PET/CT (**right**): the lesion shows a very high expression of somatostatin receptors, but is negative on (F-18) FDG PET/CT.

**Figure 6 biomedicines-12-00379-f006:**
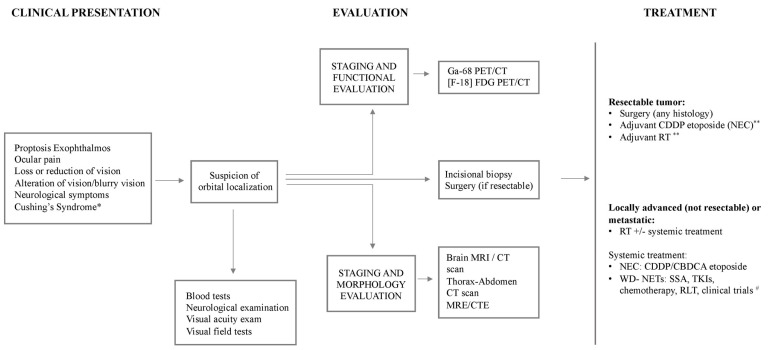
Diagnostic algorithm in the suspicion of primary orbital NENs. Abbreviations: PET/CT: Positron emission tomography/computed tomography; MRI: Magnetic Resonance Imaging; MRE: MR enterography; CTE: CT enterography (CTE); CDDP: cisplatin; CBDCA: carboplatin; NEC: neuroendocrine carcinoma; WD-NETs: well-differentiated neuroendocrine tumors; SSA: somatostatin analogues; TKIs: Tyrosine kinase inhibitors; RLT: radioligand therapy. * in the presence of diarrhea, flushing and/or bronchospasm. ** adjuvant treatment should be considered according to histology and risk factors (low-level evidence). # on the basis of available data in the literature, see GEP-NETs therapeutic algorithm.

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
