# Peer review of "Diagnostic and Therapeutic Management of Primary Orbital Neuroendocrine Tumors (NETs): Systematic Literature Review and Clinical Case Presentation"

_biomedicines, 2024, doi:10.3390/biomedicines12020379_

Round 1

Reviewer 1 Report

Comments and Suggestions for Authors

When you present the case, in the material and methods, at the immunohistochemically stain you mentioned CD56 positivity, but in figure 3 is chromogranin A positive - is not the same, please clarify this. It will be better to see the status of somatostatin receptor status (SSTR2, SSTR5, somatostatin) in your case by immunohistochemistry.

Reviewer 2 Report

Comments and Suggestions for Authors

Primary orbital neuroendocrine tumors (pO-NENs: Carcinoids, WDNETs, and NECs) are exceptionally reported.

In a systematic review of the literature (from 1966 to September 2023), the authors describe characteristics of 15 reported pO-NENs, including Carcinoids (n=5), WD-NETs (n=1), and NEC (n=9).

Additionally, the authors describe a new pO-NEN (NETG2) and propose a diagnostic algorithm in the suspicion of primary orbital NENs (including staging, morphological and functional evaluation, and “only” surgery if resectable).

The paper may be interesting for readers if the authors decide to clarify some issues, namely:

1-      State dimensions of the tumor as standard evaluation.

2-      Clarify and state expansive and or infiltrative growth of the tumor and invasion of adjacent structures.

3-      State mitotic index of the tumor (e.g., Mit. /2mm2); necrosis; lympho-vascular and perineural invasion assessments.

4-      Clarify and briefly discuss the use of G2 grade in this setting (orbit)

5-      Clarify the stage of the tumor (localized/locoregional) using largest dimension cut-offs (2 cm, >2 cm, and < 4 cm, >4 cm)? and local invasion features?

6-      Present new (better) H&E and IHQ pictures, namely for Ki-67.

7-      Discuss briefly the putative meaning of CDX2 expression in this setting.

8- Improve the proposed algorithm to include WDNETs and NECs in this setting (orbit) with other therapeutic alternatives as in the present reported case.

Comments on the Quality of English Language

Primary orbital neuroendocrine tumors (pO-NENs: Carcinoids, WDNETs, and NECs) are exceptionally reported.

In a systematic review of the literature (from 1966 to September 2023), the authors summarize characteristics of 15 reported pO-NENs, including Carcinoids (n=5), WD-NETs (n=1), and NEC (n=9).

Additionally, the authors describe a new pO-NEN (NETG2) and propose a diagnostic algorithm in the suspicion of primary orbital NENs (including staging, morphological and functional evaluation, and “only” surgery if resectable).

The paper may be interesting for readers if the authors decide to clarify some issues, namely:

1-      State dimensions of the tumor as standard evaluation.

2-      Clarify and state expansive and or infiltrative growth of the tumor and invasion of adjacent structures.

3-      State mitotic index of the tumor (e.g., Mit. /2mm2); necrosis; lympho-vascular and perineural invasion assessments.

4-      Clarify G2 grade in this setting (orbit)

5-      Clarify in the case setting (orbit) the stage of the tumor (localized/locoregional) using largest dimension cut-offs (2 cm, >2 cm, and < 4 cm, >4 cm)? and local invasion features?

6-      Present new (better) H&E and IHQ pictures, namely for Ki-67.

7-      Discuss briefly the putative meaning of CDX2 expression in this setting.

8- Clarify the proposed algorithm to include WDNETs and NECs in this setting (orbit) and other non-surgical therapeutic alternatives, as reported.

Round 2

Reviewer 2 Report

Comments and Suggestions for Authors

The revised version of the paper, with additional English editing, seems attractive to readers.